# Importance of Dark Septate Endophytes in Agriculture in the Face of Climate Change

**DOI:** 10.3390/jof10050329

**Published:** 2024-04-30

**Authors:** Victoria Huertas, Alicia Jiménez, Fernando Diánez, Rabab Chelhaoui, Mila Santos

**Affiliations:** Departamento de Agronomía, Escuela Superior de Ingeniería, Universidad de Almería, 04120 Almería, Spain; vhn616@ual.es (V.H.); alimdvilla@gmail.com (A.J.); fdianez@ual.es (F.D.); rababchelhaoui@gmail.com (R.C.)

**Keywords:** DSEs, climate change, drought, salinity, fertilisation

## Abstract

Climate change is a notable challenge for agriculture as it affects crop productivity and yield. Increases in droughts, salinity, and soil degradation are some of the major consequences of climate change. The use of microorganisms has emerged as an alternative to mitigate the effects of climate change. Among these microorganisms, dark septate endophytes (DSEs) have garnered increasing attention in recent years. Dark septate endophytes have shown a capacity for mitigating and reducing the harmful effects of climate change in agriculture, such as salinity, drought, and the reduced nutrient availability in the soil. Various studies show that their association with plants helps to reduce the harmful effects of abiotic stresses and increases the nutrient availability, enabling the plants to thrive under adverse conditions. In this study, the effect of DSEs and the underlying mechanisms that help plants to develop a higher tolerance to climate change were reviewed.

## 1. Introduction

The detrimental effects of climate change have caused long and severe periods of drought [1] in which extensive farming in the Mediterranean region has become almost impossible, and, in the case of intensive farming, the water availability has been severely reduced. The interactions between microorganisms and plants have also been affected by droughts and soil warming, causing a decrease in the symbiotic relationships between both in various ecosystems [2,3]. The endophytic relationships of fungi with plants are being widely studied to elucidate their interactions with the hosts, the type of relationship they establish, and the potential effects of such interactions. Dark septate endophytes (DSEs) (Figure 1) are conidial or sterile septate fungal endophytes, usually isolated from healthy plants, that form melanised structures, including inter- and intracellular hyphae and microsclerotia in the roots. They show low host specificity and cover a wide geographical range [4]. The presence of DSEs in environments with strong abiotic stress caused by drought [5,6,7], high salinity [8,9,10], or the presence of heavy metals [11,12,13,14,15,16], among others, is crucial to ensure plant survival [17]. This type of fungi is less affected by long periods of drought, and its colonisation rate is not reduced under such conditions. DSEs do not provide as many benefits as mycorrhizal fungi [18]; however, they show a higher colonisation rate under the conditions of abiotic stress. Despite this, the DSE colonisation in plants decreases under the simultaneous occurrence of high temperature and drought; however, it does not lead to diversity loss [2]. Most of the studies on the interaction between DSEs and plants have been conducted in forest systems [2,19,20,21,22,23].

However, the incorporation of DSEs in agriculture has become more relevant in the scientific realm, particularly owing to their increased tolerance under stress conditions [12], although studies on horticultural or extensive farming are scarce.

For example, Andrade-Linares et al. [24] obtained a higher yield and quality in tomato fruits after applying two DSEs, and Fakhro et al. [25] obtained higher morphological parameter numbers in tomato plants with the use of DSEs. Osuna-Avila and Barrow [26] described the increase in the number and length of *Daucus carota* roots through the application of *Aspergillus ustus*. However, from a commercial standpoint, a DSE-microsclerotia-based product is not available in the market. These examples will be addressed throughout this review. Recently, a patent of *Rutstroemia calopus* has been described as a DSE capable of promoting and increasing crop growth and development, even under hydric and saline stress conditions. This biostimulant effect is also visible when there is an 80% reduction in the fertilisation of cucumber plants. Moreover, studies have reported a 33.8% increase in the leaf area of plants after applying *Rutstroemia calopus* CG11560 with the reduction in fertilisation. Similarly, the total dry weight was observed to increase by 30.43% with the application of *Rutstroemia calopus* when compared with that of the standard fertilisation methods [27]. Most of the benefits associated with DSEs are related to plant biostimulation, which improves the morphological parameters [12], stimulates the plant resistance to drought, increases the secondary metabolism activity [28], improves the water, nutrient, and carbon absorption [12], increases the antioxidant enzyme activity, and enhances the development of adaptation strategies against heavy metals, among other functions [29,30]. In addition, DSE colonisation could induce a series of changes in the cell metabolism, biosynthesis, and signal pathways, thereby modulating the plant growth [31,32,33]. Owing to the properties of endophyte fungi, the most important benefits of which are obtained under abiotic stress conditions, the irrigation doses and the use of fertilisers may be reduced. Although this may not ensure a production increase, it could reduce the use of products to obtain a higher yield.

## 2. DSEs and Their Connection to Drought and Salinity Mitigation

Drought and salinity are stressful environmental factors that negatively affect plant growth by causing reactive oxygen damage [34]. Additionally, they have an impact on the hydric potential, nutrient absorption, enzyme activity, and photosynthetic pigment content. Different studies have found that, among other mechanisms, DSEs can increase the plant resistance by upregulating the antioxidant enzymes, especially the superoxide dismutase (SOD) activity, which is an important protective enzyme against reactive oxygen species (ROS) [35], increasing the polysaccharide production, and increasing the production of glutathione, proline, soluble sugar, and a large amount of melanin under stress [36] (Table 1). Likewise, endophytic fungi like *S. indica* have frequently been reported to mediate the plant root morphology and alter the composition of the root exudates [37,38,39]. Thus, DSEs and their extracellular metabolites have effects on the glycerophospholipid metabolism and the N-glycan biosynthesis pathway in the plant [40]. Hormones play a vital role in plant growth, development, and the ability to adapt to adversity, and all these processes are related to one another (Figure 2). Thus, polysaccharides are capable of retaining liquids (Figure 3) through the formation of biofilms [41].

The treatment with exopolysaccharide increases the levels of abscisic acid (ABA), promotes stomatal closure, and minimises the water loss [64] by acting as an antiperspirant agent [65]. Similarly, cellular stability occurs owing to the accumulation of the rigid and soluble osmolytes in water, which increases the osmotic pressure and causes minimal water loss. Galactose-rich heteropolysaccharide (GRH) induces the activation of the biochemical cascade necessary to maintain the hydric balance and increase the antioxidant defence [64,65,66]. Thus, the use of GRH in rice crops improved the superoxide dismutase (SOD), peroxidase (POD), and catalase (CAT) activity levels while also reducing the malondialdehyde (MDA) content, which indicates the level of lipid oxidation in the membrane. Variations in the levels of proline and soluble sugar were also observed, which improved the levels of stress-tolerant enzymes. Thus, the plants could adapt to the drought conditions owing to the role of proline as an osmoregulatory agent [64]. Similarly, the inoculation with DSEs *Phialophora* sp., *Knufia* sp., *Leptosphaeria* sp., and *Embellisia chlamydospora* in *Hedysarum scoparium* under hydric stress conditions altered the enzyme and antioxidant activity and increased the SOD and CAT levels [5], which enabled N and P absorption. Other studies on crops, such as cowpea, soybean, and rice, revealed that the DSE responses regarding the biomass depended on the saline concentration [63,67,68,69]. This property of DSEs may be attributed to their ability to produce hormones such as indole acetic acid (IAA) and gibberellins [70,71]. Thus, the application of *Alternaria alternata*, *Paraphoma pye*, and *Paraphoma radicina* to wheat crops increased the auxin levels, which can be associated with the ability of DSEs to synthetise auxins for the reduction in water intake under stress conditions and activate the expression of hormone-regulated genes [50]. Higher accumulations of SOD were also observed in wheat, rice [50,72], and citric plants [73] inoculated with *Penicillium citrinum*, *Aureobasidium pullulans*, and *Dothideomycetes* sp., both individually and combined. Moreover, the lower MDA caused by increases in the glutathione and proline contents resulted in increased drought tolerance in the plants. Zhang et al. [7] obtained similar results after inoculating sorgo with the DSE *Exophiala pisciphila*. In this case, an increase in the synthesis of the metabolites related to the secondary metabolism was also observed. This resulted in an improved hydric state owing to the opening of the stomata, which improved the transpiration rate and stomatal conductance. Similarly, the application of *Neocamarosporium phragmitis*, *Alternaria chlamydospora*, and *Microascus alveolaris* to *Lycium ruthenicum* Murr crops under drought conditions increased the glutathione content, SOD activity, and soluble protein and proline contents. Additionally, the IAA content in the plant roots also increased after inoculation with *N. phragmitis* compared with that of the control group [52]. For the first time, *Isatis indigotica* under drought stress showed structures characteristic of the DSEs *Acrocalymma vagum*, *Paraphoma chlamydocopiosa*, *Edenia gomezpomplae*, *Darksidea alpha*, *Brunneochlamydosporium nepalense*, and *Preussia terrícola* and showed increases in the IAA, proline, chlorophyll, and epigoitrin contents, which promote plant growth by improving the osmotic pressure and increasing the plant resistance to stress [47].

Moreover, DSEs have an impact on the photosynthetic activity when plants are under stress conditions. Thus, the inoculation with endophyte fungi, such as *P. indica*, *T. virens*, or *P. indica* + *T. virens*, increased the chlorophyll content and photosynthetic activity of Stevia [74]. This may be because DSEs are capable of decomposing photosynthates; DSEs enable photosynthetic feedback inhibition and improve the physiology of host plants by increasing the chlorophyll concentration and transpiration rate in stressed soils [33,75]. Furthermore, DSEs are involved in improving the ability to capture the excitation energy released by chloroplasts under drought conditions, resulting in an increased photosynthetic rate and improved leaf nutrition caused by the enhanced C assimilation [74]. Therefore, the use of DSEs *Alternaria alternata*, *Paraphoma pye*, and *Paraphoma radicina* on wheat and rice crops led to increases in the plant height, leaf growth, chlorophyll content, and photosynthetic rate, as well as a decrease in the intercellular carbon dioxide, which alleviated the damage caused to the photosynthetic processes by drought [50,72]. Conversely, the inoculation of *Ormosia hosiei* with *Acrocalymma vagum* resulted in a damage-free root cell structure and increases in the amounts of chlorophyll and carotenoids produced [45,76]. The connection between the root cells and the rhizosphere through melanised hyphae can be a strategy to survive in stressful environments and may protect plants from free radicals [77]. Similarly, the increased photosynthetic activity may be directly related to the increase in leaf area as inoculated plants have a greater capacity to retain water and show reduced evaporation [73,78,79].

The inoculation with DSEs changes the rhizosphere microbiome according to the conditions of the environment in which they are present as microorganisms are considerably specific during their life cycle. Thus, the changes in the soil characteristics caused by the DSE inoculation and water content partially explain the variations observed in the rhizosphere microbiome. The DSE inoculation under drought stress enriched the beneficial symbiotrophic fungi and growth-promoting bacteria but decreased the relative abundance of rhizosphere pathogens [48,80]. Thus, the inoculation of *Lycium ruthenicum Murr* with *A. chlamydospore* and *M. alveolaris* under drought conditions increased the general populations of arbuscular mycorrhizal fungi (AMF), fungi, bacteria, and actinomycete contents in the rhizosphere soil [52]. These results were in accordance with those reported by Li et al. [50] in *Astragalus*, wherein increases in the bacteria and beneficial fungi were observed in the rhizosphere after the inoculation with a combination of DSEs and *Trichoderma*. Therefore, the microbial richness in the rhizosphere is essential for optimal plant survival. However, not all the obtained results were related to the benefits of DSEs. The effect of DSEs depends on the type of crops they are used on, their own genotypes, and the stress conditions imposed on them [60]. These microorganisms may show negative, positive, or no effects. According to Do Santos et al. [60], in the presence of some of the microorganisms, rice crops showed beneficial or neutral results in terms of the parameters measured in their study, and negative results for some grasses under hydric stress conditions. The explanation for this phenomenon is that, in their study, two of the crops were C3 plants, whereas the other was a C4 plant, and all of them showed different adaptative mechanisms to survive under hydric stress conditions.

## 3. Implications of DSEs in Fertilisation Reduction

The excessive use of chemical fertilisers to furnish plants with the necessary nutritional requirements and improve the productivity of crops has caused serious environmental issues. The low efficiency and prolonged use of fertilisers have worsened the environmental problems [81]. Moreover, the use of intensive agricultural techniques, which particularly cause soil salinisation in arid and semiarid zones [82], along with climate change, which has reduced the amount of available water [83], have contributed to the continuous degradation of soils by lowering their quality and physiochemical properties and hindering the nutrient availability [84]. Therefore, novel techniques must be developed to help reduce the use of synthetic fertilisers and maximise the utilisation of soil nutrients in a sustainable and ecofriendly manner [85], for example, the use of soil microorganisms capable of solubilising the nutrients in the soil. Previous studies have stated that fungi are more efficient than other soil microorganisms as they solubilise nutrients that form compounds [86,87]. Notably, inside fungi, the DSEs can degrade complex substrates, secrete extracellular enzymes, such as phosphatases and cellulases, and promote nutrient absorption and use by plants by transforming them into bioavailable forms [88]. DSEs promote plant growth and aid nutrient absorption, particularly under biotic and abiotic stress conditions [89].

Phosphorus (P) is a macronutrient essential for the plant metabolism [90]. Most of the phosphorus present in the soil can be found in the form of phosphates, which form compounds together with metals, minerals (inorganic substances), and organic materials (organic substances) [91]. Although P is abundantly found in agricultural soils, most of it is present in its insoluble form; thus, it cannot be directly used by plants [92]. Consequently, to achieve an optimal crop yield, synthetic fertilisers with P are used; however, only approximately 30% of the added P can be used by crops as a large portion of the P remains immobilised in the form of insoluble phosphate or migrates in the soil [93]. Moreover, P is associated with various negative environmental impacts, such as eutrophication [94]. Several studies have indicated the capacity of DSEs to solubilise various P forms (organic and inorganic forms). Approximately 50–80% of soil organic P is present in the form of salts of oxalic acid, called phytates [95]. Mikheev et al. [96] showed that the isolate *Phialocephala fortinii* secreted phytases, which are a special group of phosphatases that catalyse phytate hydrolysis to release P [97]. Moreover, another study showed that *P. fortinii* was capable of promoting the growth of *Asparagus officinalis* using phytic acid sodium salt as the only organic source of P, which confirmed its capacity to mineralise organic P [98]. Meanwhile, other DSEs, such as *Exophiala pisciphila*, *Periconia macrospinosa*, and *Cadophora* sp., have failed to show any ability to secrete phytases [98,99]. Considering inorganic forms mainly comprising aluminium, iron, and calcium phosphates, various DSEs have shown a capacity to solubilise P both in vitro and in soil, with different levels of solubilisation in each compound according to the fungus species used [100,101,102]. This process involved phosphatase enzymes that transform insoluble P into soluble substances and included both acid and alkaline phosphatases [103].

Although their capacity for solubilising P appears to be a major contribution, some DSE strains have been found to accumulate polyphosphates [96,104]. This accumulation would enable the transport of P to plants through hyphae; this interaction has been reported in arbuscular mycorrhizal fungi (AMF) [105,106,107]. However, more studies are required to determine whether this process occurs or if these polyphosphates are directly profited by DSEs for their own metabolism.

Nevertheless, the effect of DSEs on the P absorption by plants is ambiguous [108]. Various studies have analysed the effect of the inoculation with DSEs on plant phosphorus uptake, showing mixed results based on the DSE species and studied plant.

In the case of *Lycium ruthenicum* plants, the inoculation with DSEs had a significant effect on the P found in the soil and increased the plant’s absorption of this element under hydric-deficit conditions [52]. Similarly, in pine seedlings, inoculations with DSEs contributed to the absorption of larger amounts of P [109]. Additionally, *Vaccinium macrocarpon* plants inoculated with *P. fortinii* showed an increased P content compared with plants that had not been inoculated. In particular, the aerial part and roots of the plant increased by 28 and 61%, respectively, after 10 months of cultivation [73]. Owing to the presence of the DSEs, similar results were observed in corn after inoculation with *E. pisciphila* under P-deficient and P-sufficient conditions [90,110]. In other studies, although the DSEs were capable of solubilising and mineralising the P in the soil by increasing the easily available P reserves (*Trifolium repens*), the phosphorus content was only modified when DSE and AMF were simultaneously applied [111]. In a recent study of sorgo and three DSE species (*A. alternata*, *Curvularia* sp., and *Ophiosphaerella* sp.), all three species were capable of dissolving the insoluble phosphates as in vitro reagents in the soil. However, although the inoculation with *Curvularia* sp. resulted in increases in the aerial biomass and root length of the plant, the P content of the inoculated plants showed no statistically significant differences compared with that of the control, except in the case of *A. alternata*, which had a lower P content. Therefore, greater soil P solubilisation did not result in increased absorption by the plant. A similar result was obtained in tomato plants, wherein the P content in the plants inoculated with *Cadophora* sp. and *Periconia macrospinosa* did not show a statistically significant difference compared with that of the control plants [99]. In both studies, the DSE microsclerotia was not detected in the plant roots. No endophytic structures were found in the sorgo (suggesting that the fungi maintained their saprophytic form), and the tomatoes only showed hyphae from *Cadophora* sp.

In rice, some DSE isolates were capable of increasing the P content in the aerial part of the plant compared with that in the control, whereas the P content in the other parts remained the same. Notably, such increases occurred when the plants were supplemented with an inorganic P source, suggesting that the ability of the plants to improve their absorption of P from inorganic sources depended on the species [112]. These results are also supported by another case, wherein the tomato plants inoculated with *Cadophora* sp. and *Periconia macrospinosa* showed increased growth stimulation when inorganic sources were used [99]. However, other studies suggested that DSEs are more efficient in releasing P derived from organic sources rather than that from inorganic sources [113].

Potassium (K) is an indispensable element in plant nutrition. In addition to other functions, K plays a significant role in regulating the stomatal opening and closure in leaves, and a potassium deficit can affect water–plant relations. Greater resistance to drought in the plants inoculated with DSEs could be related to a higher K content in the plant leaves [114]. Under stress conditions owing to water deficit, the K content in the roots of *Ammopiptanthus mongolicus* inoculated with two strains of *Paraconiothyrium*, *Darksidea* sp., *Embellisia chlamydospore*, and *Leptosphaeria* sp. significantly increased compared with that of the control plants. Despite the fact that the DSEs promoted the K acquisition in the roots, no statistically significant difference was observed in the quantity of K in the aerial part [57]. Meanwhile, in other studies conducted on rice, tomato, and corn, the inoculation with DSEs showed a significant increase in the K content in the plant leaves [110,115,116].

DSEs increase the bioavailability of the soil organic nitrogen (N) for the host plant as they have the ability to mineralise nitrogen [29]. They are capable of degrading nitrogenous compounds, such as proteins, ribonucleic acids, amino acids, and urea [87,88]. According to a meta-analysis by Newsham [113], this capacity represents the largest contribution of such fungi to the stimulation and improvement of plant growth. Several authors have assessed the capacity of DSEs to enable N absorption and the influence of the nitrogenous compound being provided. *Heteroconium chaetospira* is capable of promoting the growth of its natural host, namely Chinese cabbage (*Brassica rapa*), under limited nitrogenous conditions. Although the fungus was not capable of promoting the plant growth in the presence of the available N, the biomass increased to four-fold when an organic nitrogen source was used [117]. This is supported by the results of the inoculation of tomato with *P. macrospinosa*, wherein the root biomass and aerial part of the plants increased when organic N sources were used, implying that this fungus could improve the intake of N from organic sources. However, this improvement did not occur when *Cadophora* sp. was used, which shows the influence of the DSE species used on their results [99]. Another study conducted on tomatoes using an inorganic (ammonium sulphate) or organic N source (consistent in ground *Canavalia ensiformis* plants) showed that, overall, both the inoculated and non-inoculated tomatoes absorbed N more efficiently from the inorganic source than from the organic one. However, the effects of the inoculation with DSEs on N recovery was evident and significant only when the tomato plants were fertilised with *C. ensiformis* as the N accumulation in the plants was higher with this organic form (approximately 24–33%) based on the used DSE [116]. Similar results were obtained for rice, wherein the plants associated with the DSE *C. ensiformis* exhibited more efficient nitrogen intake, and the N content increased in both the aerial part and grains of the rice, resulting in increased accumulations of proteins and dry matter [118]. Another example of improved nitrogen absorption was observed in the case of *P. fortinii*, which is a DSE that promotes the growth of *Asparagus officinalis* through nitrogen mineralisation and increases the N content in the host plant [90]. The nitrogen transfer between the DSE fungi and the plant is accompanied by an exchange of carbon from the plant. Proof of this exchange was found when tracing the carbon and N in the association between *H. chaetospira* and *B. rapa*, as well as in that between tomato plants and DSEs *Pleosporales* and *Calosferiales* [116,117].

Although N mineralisation appears to be the major method through which DSEs promote N absorption, other potentially helpful mechanisms have also been suggested. For example, the DSEs in rice plants have shown higher efficiency for N intake and accumulation, resulting in an increase in the nitrogen content in the plants by 33–47% and an increase in the enzyme activity of the plasma membrane proton pump H+-adenosine triphosphatase (ATPase) [119]. This protein hydrolyses the adenosine triphosphate (ATP) in the cytosol to release protons to the cell exterior. Through the plasma membrane, hydrolysis results in an electrochemical gradient of protons necessary to regulate various physiological processes in the plant, one of them being the absorption of nutrients, such as nitrogen [120]. Additionally, an increase was observed in the vacuolar pyrophosphatase and the transcription of the OsA5 and OsA genes corresponding to the H+-ATPase isoforms. The authors suggested that this stimulation could be mediated by the metabolites secreted by DSEs [119]. In a different study, wherein rice seeds were inoculated with the DSE *pleosporales*, the absorption of NO_3_-nitrogen showed lower Michaelis–Menten constant (Km) values, indicating an increase in affinity, which was reflected in the higher N content in the aerial part of the plant [118]. Various studies have shown that the net formed by the hyphae of DSEs plays a notable role in the water and nutrient exchange between plants and soil under stress conditions, which is also dependent on the species of fungi [121,122]. For example, in a study on liquorice plants (*Glycyrrhiza uralensis*), the inoculation with DSEs increased the content of easily available N in the soil. This improvement in N availability depended on the DSE species used for the inoculation and plant irrigation system. Thus, although *Acrocalymma vagum*, *Paraphoma chrysanthemicola*, *Alternaria longissima*, *Darksidea alpha*, *Preussia terrícola*, and *Alternaria chartarum* showed increases in N when properly irrigated, under the stress conditions caused by a hydric deficit, such increases were only observed with the *Alternaria chlamydospora* and *Acremonium nepalense* species [42]. Similar results were observed in the case of *Lycium ruthenicum* [52], wherein the plants inoculated with the DSE showed the increased absorption of the available nitrogen in the soil, like that in *Triticum aestivum* L., the association of which, with a strain of *Alternaria alternata*, led to increased accumulations of nitrogen and carbon under drought-related stress [123].

DSEs can also enable the incorporation of other soil micronutrients [124]. Considering iron, when tomato seedlings or rice were inoculated with these fungi, the iron (Fe) content increased by 72–128% [116,118]. This increased Fe intake may be related to the ability of endophytic microorganisms to produce siderophores under limiting iron stress conditions [125,126,127]. Moreover, these siderophores produced by endophytes tend to have more affinity for Fe^3+^ ions than the phytosiderophores produced by plants [92]. *P. fortinii* is capable of synthetising hydroxamate siderophore, and such synthesis has been linked to the increased incorporation of Fe^3+^ by the host plant [124]. The presence of such compounds has also been confirmed by the ultra-performance liquid chromatography–mass spectrometry (UPLC-MS) analyses conducted on other DSE isolates [128]. Considering other micronutrients, the manganese and zinc contents increased in DSE-inoculated tomato and rice plants, respectively [125,126,127].

The extracellular metabolites produced by DSEs can either be directly incorporated by the plant as nutrients or act as chemical signals regulating the nutrient absorption process of plants. The use of the metabolites of *Alternaria* sp. in corn plants has been associated with a significant increase in the N content in the aerial part [129]. Conversely, in a different study conducted by Wang et al. [130] on alfalfa (*Medicago sativa*) using the same DSE species, no statistically significant differences were observed in the absorption efficiency of this element between the treated and untreated plants, although a significant improvement in the N translocation efficiency was observed. A significant increase and higher efficiency regarding the P intake were observed in both plant species, and the alfalfa plants showed a significant increase in the absorption and translocation efficiency of K. The results on the improvement in the nutrient uptake differed based on the culture time of the microorganisms prior to the metabolite extraction, emphasising the importance of this factor.

## 4. Compatibility of DSEs with Other Microorganisms

The interactions between plants and microorganisms are considerably complex owing to the presence of mutual regulation and because several factors causing their union remain unknown. Overall, studies have stated that the use of DSEs modifies the soil microbial community associated with the plant [80,109,131,132,133,134]. Thus, incorporating three DSEs (*Alternaria chlamydosporigena*, *Paraphoma chrysanthemicola*, and *Bipolaris sorokiniana*) promoted the presence of Gram-positive bacteria, Gram-negative bacteria, or the abundance of AMF in the rhizosphere of *Artemisia ordosica* based on the type of DSE and the applied saline concentrations [8]. Studies have described the natural coexistence of DSEs with other endophyte fungi and AMF in the roots of plants [135,136,137,138,139]. Despite these results, the effects of the use of DSEs on different hosts and rhizosphere microbiota remain unclear owing to the variability in the study results. This can become more complicated with the addition of new microorganisms that are used in agriculture as potential biological control agents or biostimulants. Managing these microorganisms in both extensive and intensive farming can help to reduce the use of agrochemicals and fertilisers [4,140]. However, more research is required to determine the behaviour of each DSE in various agricultural ecosystems.

Similarly, in another study, the use of endophyte fungi, DSEs, mycorrhizal fungi, or various species of *Trichoderma* played a significant role in the development of plants, even under abiotic or biotic stress conditions [48]. Thus, within the context of the current climate situation, with the frequent droughts, increasing temperature, and greater evapotranspiration of crops, microorganisms can be used to improve the intake of water; however, long periods of drought and high temperatures are also known to affect the microbiota associated with the soil and roots [141]. The soils in deserts or arid areas, where the DSEs associated with plants are abundant, are a clear example of the phenomenon [30]. Low rainfall and heat waves are associated with decreased colonisation by AMF and increased colonisation by dark septate endophytic fungi that mitigate stress when warming and drought are more severe [142,143].

The DSE management in agriculture is not as developed as in the case of *Trichoderma*; however, its use in combination with other types of microorganisms can offer multiple benefits. The co-inoculation with DSEs *Acrocalymma vagum* or *Paraboeremia putaminum* and *Trichoderma viride* improved the growth of *Astragalus mongholicus* and caused changes in the rhizosphere microbiome according to the combination of inoculants under hydric stress conditions [48,144]. The same results were obtained by Li et al. [145], who detected increases in the plant nutrients, root activity, phosphorus availability, and phosphatase activity, as well as changes in the bacterial and fungal diversity in the soil co-inoculated with *Trichoderma koningiopsis* and *Amesia nigricolor*. The co-inoculation of the genus *Darksidea* and AMF *Rhizophagus irregularis* in two plant species, namely *Artemisa tridentata* and native grass *Poa secunda*, increased the union of AMF and the formation of vesicles; however, biostimulation did not occur when the plants were not under abiotic stress [146]. The co-inoculation with the AMF *Funneliformis mosseae* and DSE *Exophiala pisciphila* improved the corn yield under high cadmium (Cd) stress, and also significantly reduced the Cd transfer from the roots to the stem, which was closely related to the changes in the photosynthesis physiology and corn roots [134]. Conversely, an interaction between both fungi was observed, which negatively impacts the colonisation rate. Thus, Deram et al. [147] detected that the colonisation by septate fungi was reduced in the presence of mycorrhizal fungi in non-polluted soils; however, it increased in polluted soils, partially owing to a reduction in mycorrhisation, which was impacted by the presence of heavy metals. Meanwhile, Li et al. [145] observed that the co-inoculation with *Trichoderma koningiopsis* and the DSE *Amesia nigricolor* reduced the colonisation rate of the DSE. Similarly, the level of mycorrhisation was not affected by the colonisation of the *Phialocephala fortinii* s.l.–*Acephala applanata* species complex (PAC), although the DSE complex colonisation reduced. This negative effect on DSE colonisation has been evidenced by other studies [148,149,150]. However, other studies have revealed positive effects showing an increase in DSE colonisation [111,151,152], or neutral effects, wherein the colonisation rates of both the co-inoculants were unaffected [111,153,154,155,156]. Table 2 lists various studies on the compatibility between DSEs and other microorganisms.

These effects are not only observed among fungal species but also in bacteria. The data suggest that some bacteria from the *Rhizobium*/*Agrobacterium* group are endophytes of some DSE species [151,170,171]. Thus, the combined use of *Agrobacterium pusense* isolated from the DSE *Veronaeopsis simplex* in tomatoes resulted in an increased number of roots colonised by the fungus. The same finding was observed by Silva et al. [158], who described that the root colonisation by AMF and DSEs can be increased by the application of *Azospirillum brasilense*. Wu et al. [172] observed a high colonisation rate of mycorrhizal fungi with DSEs in *C. korshinskii* roots, which indicated the existence of symbiotic relationships between them, as well as with *Rhizobium* under desert conditions.

Different factors have an impact on the colonisation rates of various microorganisms. Huo et al. [159] showed that the colonisation of DSEs and AMF had different responses to the climate conditions and soil types in different geographical areas, wherein the DSEs were more dependent on the abiotic stress conditions. Lugo et al. [161] found synergy between both AMF and DSEs, although with a different response based on the altitude and environmental factors, as the colonisation rate of the AMF decreased and that of the DSEs did not. Conversely, Ruotsalainen et al. [18] showed that the colonisation of AMF and DSEs depends on the season of the year, and that a certain degree of specificity was imposed by the host. Such a dependency from the host was also observed by Fernando and Currah [165], who also described a dependency on the harvesting conditions. The inoculant dose is another factor that must be considered regarding the root colonisation rate, as well as the potential beneficial effect on the plant. Xie et al. [173] observed that, the higher the inoculation dose of the DSE *Alternaria* sp. paired with a fixed dose of AMF *Diversispora epigaea*, the lower the colonisation of the DSE, indicating competition for space between both fungal species. A biostimulant effect was observed in corn plants under the same conditions [172].

## 5. Compatibility of DSEs with Active Chemical Substances

Although DSEs are indispensable for the comprehensive management of disease control and crop biostimulation, studies on the compatibility of agrochemical management with DSEs are scarce. The compatibility not only depends on the microorganisms and the active chemical substance but also on other factors, such as the amount of organic matter in the soil and the microbial community size [174]. DSEs have an increased capacity for counteracting the negative effects of heavy metals in plants owing to their high antioxidant enzyme activity [4]. Furthermore, they show increased resistance to compounds such as tannins, possibly owing to a physical mechanism and the production of tannase enzymes and polyphenol oxidase [175]. However, their resistance capacity against agrochemicals remains largely unknown.

Repeated applications of herbicides are particularly harmful to soil health and microbe–plant associations. Various studies have shown that glyphosate rapidly decreases the DSE colonisation percentage in different plant species, which depends on the dose and number of applications [176]. Spagnoletti and Chiocchio [177] showed that DSEs *Alternaria alternata* and *Cochliobolus* sp. are tolerant to glyphosate, carbendazim, and cypermethrin, which were assessed with the recommended agronomic doses.

Mancozeb is another widely used fungicide. Manalu et al. [178] isolated the DSEs obtained from chili pepper grown in an environment with pesticides and detected two isolates that were resistant to 400 ppm of mancozeb. Additionally, the presence of mancozeb in the medium resulted in changes in the colour of the mycelium. This change in colour has also been described by Wyss et al. [179], who also detected a change in the shape of the hyphae and conidia of *Phomopsis amaranthicola* in the presence of herbicides from the family of imidazolines. Widyaningsih and Triasih [180] conducted an in vitro test on the resistance of three DSEs to Propineb 70%, mancozeb 80%, and glyphosate. All the DSEs were sensitive to the different tested doses, showing a decrease in sporulation with increases in the dose of both fungicides. In the case of glyphosate, they were tolerant and only one isolate had a decreased sporulation capacity. None of the cases showed changes in the mycelium shape, although a decrease in density was observed in the Petri dishes. Conversely, tricyclazole inhibits the growth of *E. pisciphila* by inhibiting ergosterol and melanin biosynthesis; however, no toxicity in low concentrations was observed [181].

Seed treatment with systemic fungicides for controlling seed-transmitted diseases is a common practice before harvesting. However, this treatment also affects non-target organisms, such as endophytes [182], as it compromises the germination and early growth of seedlings [183], which is the case in the non-pathogenic *Alternaria* found in seeds.

## 6. Conclusions

Dark septate endophytes have shown a capacity for mitigating and reducing the harmful effects of climate change in agriculture, such as salinity, drought, and reduced nutrient availability in the soil. Therefore, their use will become a good choice for crop management in the face of the current environmental challenges, as well as an alternative that is conducive to more sustainable agriculture. However, many obstacles need to be overcome before implementing DSEs in the field. Further research is required to shed light on the mechanisms involved, interactions between the plant and other microorganisms present in the soil, and compatibility with the frequently used active chemical substances. Conversely, additional studies are essential for the development of DSE formulations at an industrial scale to be used in agriculture in the future. Although further research is still required, the interest in this field has continued to grow in recent years, and the use of microorganisms in agriculture has substantially advanced.

## Figures and Tables

**Figure 1 jof-10-00329-f001:**
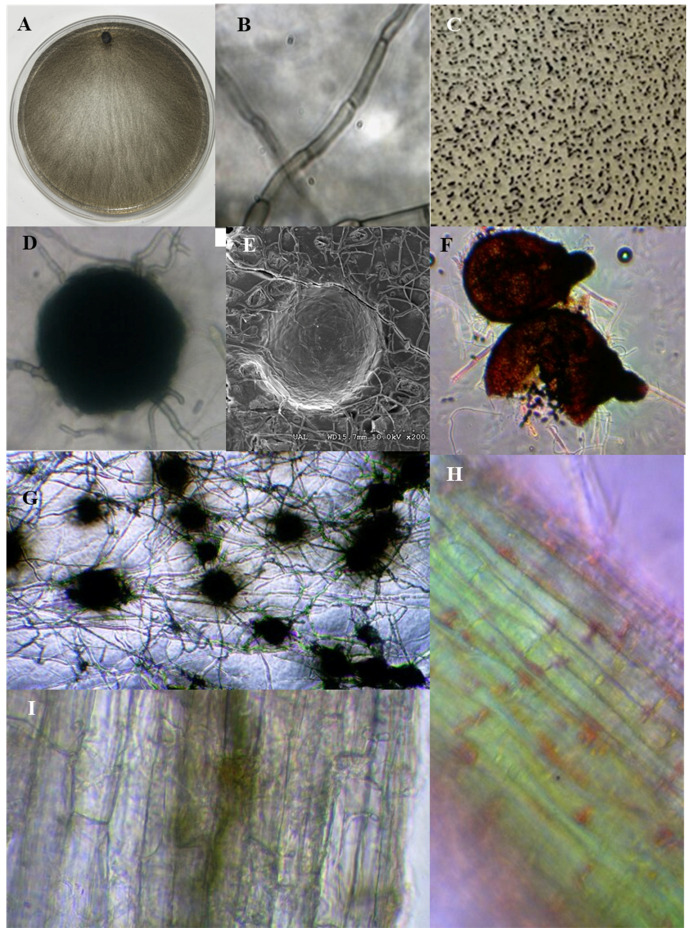
Typical characteristics of dark septate endophytes (DSEs). DSEs (*Macrophomina phaseolina*) grown in Petri dishes with PDA medium (**A**) and melanised hyphae (magnification 400×) (**B**). Microsclerotia (**C**,**G**) and microsclerotium ((**G**) magnification 100×) (**D**) on the PDA medium of *Rutstroemia calopus*. Scanning electron micrographs of microsclerotium of *Rutstroemia calopus* (**E**). *Cleistothecia* of *Sordaria* magnification 100×) (**F**). DSE root colonisation (magnification 400×) (**H**,**I**).

**Figure 2 jof-10-00329-f002:**
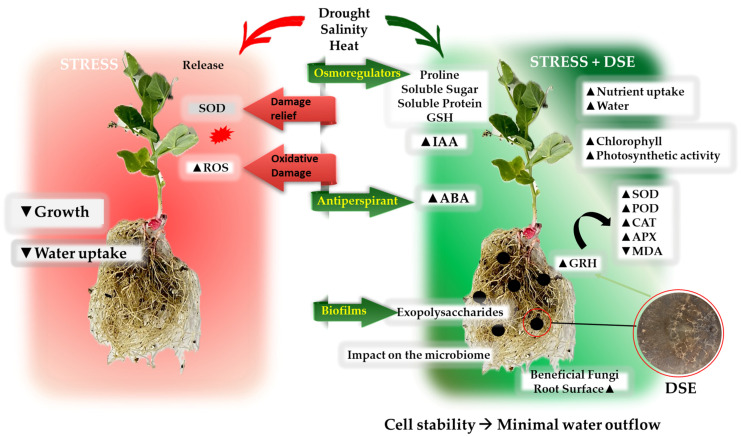
Schematic representation regarding the different effects and mechanisms of action reported with DSE inoculation in plants. The plant on the left represents a non-colonised plant subjected to drought, heat, or salinity stress, and on the right is a DSE-colonised plant subjected to stress. ▲: increase; ▼: decrease; CAT: catalase; SOD: superoxide dismutase; ROS: reactive oxygen species; POD: peroxidase; MDA: malonaldehyde; GSH: glutathione; APX: ascorbate peroxidase; IAA: indole acetic acid; ABA: abscisic acid; GRH: galactose-rich heteropolysaccharide.

**Figure 3 jof-10-00329-f003:**
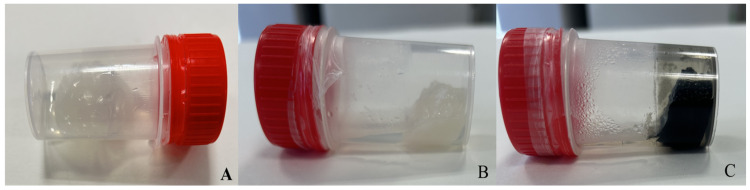
Gelatinisation of the liquid culture medium at 25 °C by *Stromatinia narcissi* (**A**,**B**) and *Macrophomina phaseolina* (**C**).

**Table 1 jof-10-00329-t001:** Effects of DSEs on plant growth and abiotic stress alleviation (drought, salinity, or heat).

DSE Species	Host Plant	Stress Type	Main Effect	Reference
*Acremonium nepalense* *Acrocalymma vagum* *Alternaria chartarum* *Alternaria chlamydospora* *Alternaria longissima* *Darksidea alpha* *Paraphoma chrysanthemicola* *Preussia terricola*	*Glycyrrhiza uralensis*	Drought	▲ Available N and organic matter in the rhizospheric soil. ▲ Glycyrrhizin content ▲▼ Osmotic stress tolerance of the eight DSE species was highly variable ▲▼ Biostimulant	[42]
*Acrocalymma aquatica* *Alternaria alstroemeriae*	*Isatis indigotica*	Drought	▲ Epigoitrin content ▲ Biostimulant	[43]
*Acrocalymma vagum* *Fusarium acuminatum* *Paraboeremia putaminum*	*Glycyrrhiza uralensis*	Drought	▲ Available N and organic matter in the rhizospheric soil. ▲ Glycyrrhizin content ▲ Biostimulant (according to DSE) Impact on the microbiome ▲ Photosynthesis rate, ▲ Stomatal conductance ▲ Intercellular CO_2_ concentration ▲▼ CAT, POD, and MDA (according to DSE)	[44]
*Acrocalymma vagum*	*Ormosia hosiei*	Drought	Changes in root morphology ▲▼ Balance of endogenous hormones	[45]
*Acrocalymma vagum* *Alternaria chartarum* *Paraphoma chrysanthemicola*	*Artemisia ordosica*	Drought	▲ Biostimulant ▲ SOD ▲ Soluble Proteins content ▲ GSH ▲ Endogenous hormones ▼ MDA ▲ Chlorophyll content ▲ Proline content	[46]
*Acrocalymma vagum* *Edenia gomezpompae* *Darksidea alpha*	*Isatis indigotica*	Drought	▲ Epigoitrin content Biostimulant ▲ Soluble Proteins content ▲ SOD ▲ Proline content	[47]
*Acrocalymma vagum* *Paraboeremia putaminum* *(with non DSE T. viride)*	*Astragalus mongholicus*	Drought	Stronger impact on the microbiome ▲ Biostimulant	[48]
*Acrocalymma vagum* *Paraboeremia putaminum*	*Glycyrrhiza uralensis*	Drought	Stronger impact on the microbiome Biostimulant	[49]
*Alternaria alternata* *Macrophomina pseudophaseolina*	*Astragalus mongholicus*	Drought	▲ SOD ▲ Endogenous hormones ▲ Soluble Proteins content Biostimulant ▲GSH ▲ Calycosin-7-O-β-D-glucoside content ▲ Formononetin content	[50]
*Alternaria alternata* *Paraphoma pye* *Paraphoma radicina*	*Triticum aestivum*		▲ SOD ▲ Soluble Proteins content Biostimulant ▲ GSH ▲ Photosynthetic efficiency ▲ Chlorophyll content ▼ MDA ▲ Soluble sugars content	[51]
*Alternaria chlamydosporigena* *Bipolaris sorokiniana* *Paraphoma chrysanthemicola*	*Artemisia ordosica*	Salinity	▲ Biostimulant ▲ GSH ▲ SOD ▲ IAA content Stronger impact on the microbiome	[8]
*Alternaria chlamydospore* *Microascus aleolaris* *Neocamarosporium phragmitis*	*Lycium ruthenicum*	Drought	▲ Biostimulant Stronger impact on the microbiome	[52]
*Alternaria* sp.	*Astragalus mongholicus*	Heat	▲ Photosynthetic efficiency ▲ Leaf circumference and area ▲ Stomatal opening density ▲ Chlorophyll content	[53]
*Anteaglonium gordoniae*	*Vaccinium corymbosum*	Drought	▲ Biostimulant Changes in cell metabolism and biosynthesis and signal pathways	[33]
*Anteaglonium gordoniae*	*Vaccinium corymbosum*	Salinity	▲ Biostimulant ▲ APX, CAT, and POD ▼ MDA and H_2_O_2_ ▲ Vitamin C ▲ upregulated transcription factor *Vab*ZIP12 (oxidative stress-related genes were regulated by *Vab*ZIP12)	[54]
*Cadophora* sp. *Leptodontidium* sp. *P. macrospinosa*	*Solanum lycopersicum*	Salinity	▲ Nutrient uptake (P and N) ▲ Biostimulant ▲ Soluble Proteins content	[55]
*Curvularia* sp.	*Populus tomentosa*	Salinity	▲ Biostimulant ▲ SOD and APX ▲ Chlorophyll a,b ▲ Proline contents	[56]
*Darksidea* sp. *Embellisia chlamydospora* *Knufa* sp. *Leptosphaeria* sp. *Phialophora* sp.	*Astragalus mongholicus*	Drought	▲ Biostimulant ▲ Nutrient K, Mg, and Ca	[57]
DSE	*Astragalus mongholicus*	Heat	▼ MDA ▲ SOD and POD ▲ GSH and putrescine (Put) ▲ Soluble sugars and proline content.	[58]
DSE	*Spartina alterniflora*	Salinity	▲ Biostimulant	[59]
DSE	*Oryza sativa*	Drought	▲Δ Biostimulant ▲ APX, CAT (differential responses of genotypes to inoculation)	[60]
*Embellisia chlamydospora* *Knufia* sp. *Leptosphaeria* sp. *Phialophora* sp.	*Hedysarum scoparium*	Drought	Biostimulant ▲ SOD and CAT	[5]
*Exophiala pisciphila*	*Sorghum bicolor*	Drought	▲ Biostimulant ▲ Photosynthesis rate, ▲ Stomatal conductance ▲ Intercellular CO_2_ concentration ▲ CAD, PAL, G-POD ▲ Secondary metabolites	[7]
*Exophiala* sp.	*Coelogyne viscosa*	Drought	Biostimulant ▲ SOD and CAT ▼ POD ▲ Soluble Proteins content ▲ Soluble sugars and proline content.	[61]
*Lulwoana* sp.	*Spartina alterniflora*	Salinity	▲▼ Biostimulant (differential responses of *S. alterniflora* genotypes to inoculation)	[62]
*Melanconiella elegans* *Sordariomycetes* sp.-B′2	*Vigna unguiculata*	Salinity	Biostimulant ▲ Leaf circumference and area ▲ Stomatal opening density ▲ Chlorophyll content, transpiration rate, and net photosynthetic rate	[63]

Note: ▲: increase; ▼: decrease; Δ: null effect; SOD: superoxide dismutase; CAT: catalase; POD: peroxidase; MDA: malonaldehyde; GSH: glutathione; bZIP: basic leucine zipper; H_2_O_2_: hydrogen peroxide; APX: ascorbate peroxidase; IAA: indole acetic acid; CAD: cinnamyl alcohol dehydrogenase; PAL: phenylalanine ammonia-lyase; G-POD: guaiacol peroxidase.

**Table 2 jof-10-00329-t002:** Studies on inoculant compatibility between DSEs and other microorganisms.

DSE Species	Co-Inoculum	Host Plant	Reference
*Acrocalymma vagum* *Paraboeremia putaminum*	*Trichoderma viride*	*Astragalus mongholicus*	[48]
*Alternaria* sp.	*Diversispora epigaea*	*Zea mays*	[157]
*Amnesia nigricolor*	*Trichoderma koningiopsis*	*Vaccinium corymbosum*	[145]
*Cadophora* sp.	*Funneliformis mosseae*	*Lolium perenne*	[149]
*Cladosporium cladosporioides*	*Oidiodendron citrinum*	*Vaccinium corymbosum*	[150]
*Darksidea*	*Rhizophagus irregularis*	*Artemisa tridentata* *Poa secunda*	[146]
DSE	*Azospirillum brasilense* AMF	*Zea mays*	[158]
DSE	AMF	*Medicago sativa*	[152]
DSE	AM	*Artemisia* sp.	[159]
DSE	AM	*Parkia timoriana* *Solanum betaceum*	[160]
DSE	AM	*Poaceae*	[161]
DSE	AM	*Alchemilla glomerulans Carex vaginata* *Ranunculus acris* ssp. *pumilus* *Trollius europaeus*	[162]
DSE	AMF	*Polygonatum kingianum*	[163]
DSE	*Rhizoglomus clarum* *Claroideoglomus etunicatum Acaulospora morrowiae*	*Paspallum millegrana*	[156]
*Exophiala pisciphila*	*Funneliformis mosseae*	*Zea mays*	[110,134]
*Exophiala pisciphila*	*Rhizophagus irregularis*	*Zea mays*	[164]
*Phialocephala fortinii*	*Glomus intraradices*	*Medicago sativa*	[122]
*Phialocephala fortinii*	*Leptodontidium orchidicola*	*Potentilla fruticosa* *Dryas octopetala* *Salix glauca* *Picea glauca*	[165]
*Phoma leveillei*	*Acaulospora laevis*	*Cucumis sativus*	[166]
*Macrophomina pseudophaseolina* *Paraphoma radicina*	*T. afroharzianum*, *T. longibrachiatum*	*Astragalus mongholicus*	[50]
*Paraphoma chrysanthemicola* *Gaeumannomyces cylindrosporus*	*Suillys bovinus* *Amanita vaginata*	*Pinus tabulaeformis*	[19]
*Phialocephala fortinii*	*Suillus bovinus*	*Pinus massoniana*	[144]
*Phialocephala turiciensis* *Acephala applanata* *P. glacialis* *Phaeomollisia piceae*	*Gigaspora rosea*	*Trifolium repens*	[111]
*Phialocephala fortinii* *Phialocephala subalpina*	*Laccaria bicolor*	*Pseudotsuga menziesii*	[150]
*Phialocephala fortinii s.l.–Acephala applanata species complex (PAC)*	*Hebeloma crustuliniforme*	*Picea abies*	[167]
*Piriformospora indica*	*Glomus mosseae*	*Triticum aestivum*	[168]
*Piriformospora indica*	*T. harzianum*	*Piper nigrum*	[169]
*Cladosporium cladosporioides* *Exophiala salmonis* *Phialophora mustea* *Paraphoma chrysanthemicola Gaeumannomyces cylindrosporus*	*Schizophyllum* sp. *Suillus laricinus* *Amanita vaginata* *Handkea utriformis* *Suillus tomentosus* *Suillus bovinus* *Suillus lactifluus*	*Pinus tabulaeformis*	[20]
*Acephala applanate* *Phialocephala europaea* *Phialocephala fortinii* *Phialocephala Helvetica* *Phialocephala letzii* *Phialocephala subalpina* *Phialocephala turiciensis* *Phialocephala uotolensis* *Acephala macrosclerotiorum* *Phialocephala glacialis*	*Paxillus involutus* *Rhizoscyphus ericae*	*Vaccinium corymbosum*	[21]
*Veronaeopsis simplex*	*Agrobacterium pusense*	*Solanum lycopersicum*	[151]

## Data Availability

Not applicable.

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
