# Peer review of "Importance of Dark Septate Endophytes in Agriculture in the Face of Climate Change"

_jof, 2024, doi:10.3390/jof10050329_

Round 1

Reviewer 1 Report

I have had the opportunity to thoroughly review this manuscript submitted for publication in the Journal of Fungi. The manuscript delves into the symbiotic relationships between DSEs and plants, highlighting how these endophytes improve plant resilience to adverse environmental conditions. It systematically reviews existing literature to discuss the mechanisms through which DSEs confer stress tolerance, including physiological and biochemical pathways. The authors also compare the efficacy of DSEs with that of other symbiotic organisms like mycorrhizal fungi, offering a nuanced understanding of the potential advantages of DSEs in agricultural practices.

The manuscript is well-structured, presenting a logical flow of ideas and a clear argumentation line. The authors have done an excellent job of synthesizing a vast amount of research to provide a coherent overview of the current state of knowledge on DSEs in agriculture. The review is comprehensive, covering various aspects of DSE-plant interactions and their implications for agriculture under climate change scenarios. However, there are several minor suggestions for enhancing the readability of the manuscript.

1. On Line 76, the manuscript states that DEGs can enhance the "quality" of plant exudates. This assertion is problematic because determining the beneficial or detrimental nature of the quality of root exudates is challenging. This difficulty arises from the uncertainty regarding which organisms might benefit from alterations in the composition and/or quantity of plant exudates under varying abiotic stress conditions. The sentence should be revised for clarity.

2. In the text spanning lines 78 to 80, the connection indicating a cause-and-effect relationship between the two sentences is not apparent. Please either clarify how these sentences are related in terms of causality or restructure them to better convey their intended relationship.

3. The content within lines 70 to 92 appears unrelated to DSEs or their functions. Please review and possibly revise or remove these sentences to ensure the manuscript maintains a focused discussion on DSEs and their roles.

4. In lines 102 to 104, it is observed that two different styles of referencing are used. Please standardize the reference style throughout the manuscript.

5. Lines 92 - 120. It would be beneficial to either summarize these findings in a conclusion that highlights their significance or to provide a detailed explanation of Figure 2, which presumably illustrates these concepts visually.

6. Line 153. The phrase "increased fungi and bacteria contents" requires clarification. It is necessary to specify whether the inoculation with A. chlamydospore and M. alveolaris leads to a general increase in the fungal and bacterial populations within the rhizosphere or if it selectively enhances certain populations. This clarification is also applicable to the similar statement found in line 155.

7. On Line 241, please revise the writings of P content to ensure consistency throughout the manuscript.

Overall, the manuscript provides a valuable contribution to the field of agricultural sciences. With minor revisions, this manuscript could be an excellent resource for potential readers.

Author Response

RESPONSE TO REVIEW 1

Thank you very much for all the recommendations provided. The manuscript has been profoundly changed and the suggestions have been incorporated.

  1. On Line 76, the manuscript states that DEGs can enhance the "quality" of plant exudates. This assertion is problematic because determining the beneficial or detrimental nature of the quality of root exudates is challenging. This difficulty arises from the uncertainty regarding which organisms might benefit from alterations in the composition and/or quantity of plant exudates under varying abiotic stress conditions. The sentence should be revised for clarity.

The sentence has been modified: DSE can increase plant resistance by up-regulating antioxidant enzymes, especially superoxide dismutase (SOD) activity, namely, an important protective enzyme against reactive oxygen species (ROS) [31], increasing polysaccharide production, and increasing the production of glutathione, proline, soluble sugar, and a large amount of melanin under stress [32]. Likewise, endophytic fungi like S. indica has been frequently reported to mediate plant root morphology and alter composition of the root exudates.

Two new references have been introduced:

Hosseini, F.; Mosaddeghi, M.R. Effects of Serendipita indica inoculation of four wheat cultivars on hydraulic properties and aggregate stability of a calcareous soil. Plant Soil 2021,469,347-367.

Olanrewaju, O.S.; Ayangbenro, A.S.; Glick, B.R.; Babalola, O.O. Plant health: feedback effect of root exudates-rhizobiome interactions. Appl. Microbiol. Biotechnol. 2019, 103(3):1155-1166.

  1. In the text spanning lines 78 to 80, the connection indicating a cause-and-effect relationship between the two sentences is not apparent. Please either clarify how these sentences are related in terms of causality or restructure them to better convey their intended relationship.

The sentence has been modified: Hormones play a vital role in plant growth, development, and the ability to adapt to adversities, and all these processes are related to one another (Figure 2). Thus, polysaccharides are capable for retaining liquids (Figure 3) through the formation of biofilms [33].

  1. The content within lines 70 to 92 appears unrelated to DSEs or their functions. Please review and possibly revise or remove these sentences to ensure the manuscript maintains a focused discussion on DSEs and their roles.

The sentence:  Among other mechanisms, different studies have found that DSE can increase plant resistance by up-regulating of antioxidant enzymes, especially superoxide dismutase (SOD) activity, an important protective enzyme against reactive oxygen species (ROS) [31], as well as improving the quality………. all the information explained refers to DSE. English has been improved to make the sentences make more sense.

  1. In lines 102 to 104, it is observed that two different styles of referencing are used. Please standardize the reference style throughout the manuscript.

The error has been removed.

  1. Lines 92 - 120. It would be beneficial to either summarize these findings in a conclusion that highlights their significance or to provide a detailed explanation of Figure 2, which presumably illustrates these concepts visually.

An explanation has been included in the legend of figure 2 and the different functions have been incorporated in table 1.

  1. Line 153. The phrase "increased fungi and bacteria contents" requires clarification. It is necessary to specify whether the inoculation with A. chlamydospore and M. alveolaris leads to a general increase in the fungal and bacterial populations within the rhizosphere or if it selectively enhances certain populations. This clarification is also applicable to the similar statement found in line 155.

According to the bibliographic references, there is an increase in general populations of fungi and bacteria, without mentioning specific genera. In others, there are tendencies of increase of some species over others, depending on the type of soil.

The sentence has been modified: Inoculation with DSEs changes the rhizosphere microbiome according to the conditions of the environment in which they are present, as microorganisms are considerably specific during their life cycle. Thus, Changes in soil characterisitics caused by dse inoculation and water content partially explain the variations observed in the rhizosphere microbiome. DSE inoculation under drought stress enriched beneficial symbiotrophic fungi and growth-promoting bacteria, but decreased the relative abundance of rhizosphere pathogens.58,59. Thus, inoculation of Lycium ruthenicum Murr with A. chlamydospore and M. alveolaris under drought conditions increased the general populations arbuscular mycorrhizal fungi, fungi, bacteria, and actinomycete contents in the rhizosphere soil [48].

  1. On Line 241, please revise the writings of P content to ensure consistency throughout the manuscript.

The sentence has been modified

Reviewer 2 Report

The title is consistent with the study conducted.

The abstract is concise and relevant.

The figures presented in this manuscript are qualitative but insufficient. It is recommended to introduce new figures:

Examples: 

- of how DSEF endophytes colonize the plant,

- a general overview of plant response to abiotic stress,

- mechanisms of action of DSEF endophytes with other microorganisms,

- mechanism of action of septate endophytes in reducing fertilization."

The work has a medium degree of originality, with similar reviews having been carried out previously.

The title is consistent with the study conducted.

The abstract is concise and relevant.

The figures presented in this manuscript are qualitative but insufficient. It is recommended to introduce new figures:

Examples: 

- of how DSEF endophytes colonize the plant,

- a general overview of plant response to abiotic stress,

- mechanisms of action of DSEF endophytes with other microorganisms,

- mechanism of action of septate endophytes in reducing fertilization."

The work has a medium degree of originality, with similar reviews having been carried out previously.

 For the improvement of the manuscript, I recommend consulting the book chapter: Endophytic Fungi and Their Symbiotic Potential as a Sustainable Agriculture Crop Growth Promoter under Biotic and Abiotic Stress Conditions, Surono,  Hamim, Jamaludin Malik, and Nicho Nurdebyandaru, Dark Septate

For the improvement of the manuscript, I recommend consulting the book chapter: Endophytic Fungi and Their Symbiotic Potential as a Sustainable Agriculture Crop Growth Promoter under Biotic and Abiotic Stress Conditions, Surono,  Hamim, Jamaludin Malik, and Nicho Nurdebyandaru, Dark Septate

Author Response

RESPONSE TO REVIEW 2

 Thank you very much for all the recommendations provided. The manuscript has been profoundly changed and the suggestions have been incorporated.

The figures presented in this manuscript are qualitative but insufficient. It is recommended to introduce new figures:

Examples: 

- of how DSEF endophytes colonize the plant,

- a general overview of plant response to abiotic stress,

- mechanisms of action of DSEF endophytes with other microorganisms,

- mechanism of action of septate endophytes in reducing fertilization."

The work has a medium degree of originality, with similar reviews having been carried out previously.

 For the improvement of the manuscript, I recommend consulting the book chapter: Endophytic Fungi and Their Symbiotic Potential as a Sustainable Agriculture Crop Growth Promoter under Biotic and Abiotic Stress Conditions, Surono,  Hamim, Jamaludin Malik, and Nicho Nurdebyandaru, Dark Septate

The figure has been modified to include more information. The book chapter has been consulted and included in the references.

Reviewer 3 Report

This publication presents a review on the dark septate endophytes and their potential use in agriculture to mitigate climate change related stresses. The adopted approach in this review was interesting since DSEs figure among the significant components of the plant microbiome and could be a promising asset to boost plant tolerance to different stresses.

The manuscript was well introduced, and the authors shed lights on the mechanisms orchestrating the DSEs’ role in the alleviation of climate change induced stress on crops. However, the manuscript needs major revisions to be suitable for publication in Journal of Fungi.

General comments

- The English of this manuscript needs moderate improvement (grammatical and punctuation checks).

- Please provide some examples of (up/downregulated) genes involved in the improvement of plant tolerance to environmental stresses when inoculated with DSEs. Please include examples in each subsection dealing with this aspect.

- Please provide, at the end of each section, the new avenues to develop regarding the discussed aspect.

- Please provide a subsection on the role of DSEs in mitigating heat stress since it is a climate change induced stress. Also, provide more examples of salinity stress since the main reported stress in this review in drought stress.

- Please provide a more detailed figure instead of Figure 2 including the multiple mechanisms underlying the effect of DSEs on plants under climate change related stresses.

- Please provide a table before Table 1 and similar to this one with the following columns: Stress type/DSE species/host plant/Main effects/ Ref.

Other comments

- Please further develop your abstract and provide more details on the aim of the review at the end.

1. L17: please avoid using abbreviation as keywords

2. L20: Please change “long, severe” to “long and severe”.

3. Figure 1: please provide a scale bar in all subfigures

4. L50: Please change “tomato plants. While” to “tomato plants, while”

5. L73-78: long sentence, please shorten or divide into two sentences

6. L80: Please reformulate, here we cannot start the sentence with “Thus”.

7. L87-88: Please change “superoxide dismutase (SOD), peroxidase activity (POD) and catalase activity (CAT)” to “SOD, peroxidase (POD) activity and catalase (CAT) activity”.

8. L91: “which improved the number of stress-tolerant enzymes.”, do you mean “the activity” instead of “the number”?

9. L96: Please provide the significance of the abbreviations (N, P) at the first appearance in the text. Please check throughout the manuscript. Please use abbreviations for many repeated words, such as phosphorus, nitrogen, potassium ….

10. L102-104: Please correct the citation format.

11. L103-107: long sentence

12. L116: Please change “ under stress due to drought” to “under drought stress”.

13. Figure 2: Please provide more details on the mechanisms involved in the mitigation of climate change associated stress on plants when inoculated with DSEs. Please change “Chlorophille” to “Chlorophyll” and “activity photosynthetic” to “Photosynthetic activity”.

14. Figure 3: It is not relevant to provide a figure to confirm one statement in one sentence. I recommend removing this figure.

15. L233: “harvesting”, do you mean “cultivation”?

16. L241: “Rise”, do you mean “Rice”?

17. L362-363: “of endophyte fungi such as mycorrhizal fungi”, are they endophyte fungi?

18. L367: Please change “water available” to “available water”.

19. L383: Please provide the meaning of “HMA”, do you mean AMF? The same for “PAC” (L396).

20. Table 1: Please add another column after the “host plant” column for “the main induced effect”, and if the terms “DSE” and “AMF” mean “DSE or AMF consortium”, please specify it. In the same table why “AM” is different from “AMF”?

21. L441: please change “show” to “showed”.

Author Response

RESPONSE TO REVIEW 3

Thank you very much for all the recommendations provided. The manuscript has been profoundly changed and the suggestions have been incorporated.

General comments

- The English of this manuscript needs moderate improvement (grammatical and punctuation checks).

English has been reviewed by native English experts

- Please provide some examples of (up/downregulated) genes involved in the improvement of plant tolerance to environmental stresses when inoculated with DSEs. Please include examples in each subsection dealing with this aspect.

Gene expression studies based on the application of DSE in plants are very rare and have not been performed under abiotic stress conditions. New references and examples have been included in the text.

- Please provide, at the end of each section, the new avenues to develop regarding the discussed aspect.

The objectives of this study are to collect existing information on the benefits of DSE. In different parts of the text the need for further studies to manage these fungi in agriculture is indicated. The authors believe that the objective is not to determine new avenues to develop.

- Please provide a subsection on the role of DSEs in mitigating heat stress since it is a climate change induced stress. Also, provide more examples of salinity stress since the main reported stress in this review in drought stress.

More information has been collected in the new table. In this manuscript we want to give more importance to drought than to salinity. In fact, the examples given of salinity are mostly combined with drought stress. Some examples of temperature increase are incorporated.

- Please provide a more detailed figure instead of Figure 2 including the multiple mechanisms underlying the effect of DSEs on plants under climate change related stresses.

Figure 2 has been modified.

- Please provide a table before Table 1 and similar to this one with the following columns: Stress type/DSE species/host plant/Main effects/ Ref.

A New Table has been added

Other comments

- Please further develop your abstract and provide more details on the aim of the review at the end.

Abstract has been improved.

  1. L17: please avoid using abbreviation as keywords

eliminated

L20: Please change “long, severe” to “long and severe”.

The sentence has been modified.

Figure 1: please provide a scale bar in all subfigures.

Microscope magnification is included

 L50: Please change “tomato plants. While” to “tomato plants, while”

The sentence has been modified.

L73-78: long sentence, please shorten or divide into two sentences

The sentence has been modified.

L80: Please reformulate, here we cannot start the sentence with “Thus”.

It is correct to start sentences with THUs. In science there are many sentences that begin with thus to affirm something.

  1. L87-88: Please change “superoxide dismutase (SOD), peroxidase activity (POD) and catalase activity (CAT)” to “SOD, peroxidase (POD) activity and catalase (CAT) activity”.

The sentence has been modified.

  1. L91: “which improved the number of stress-tolerant enzymes.”, do you mean “the activity” instead of “the number”?

The “number” has been modified by “levels”.

L96: Please provide the significance of the abbreviations (N, P) at the first appearance in the text. Please check throughout the manuscript. Please use abbreviations for many repeated words, such as phosphorus, nitrogen, potassium ….

The abbreviations have been included.

  1. L102-104: Please correct the citation format.

Error has been modified

  1. L103-107: long sentence

The sentence has been modified.

  1. L116: Please change “ under stress due to drought” to “under drought stress”.

The sentence has been modified.

  1. Figure 2: Please provide more details on the mechanisms involved in the mitigation of climate change associated stress on plants when inoculated with DSEs. Please change “Chlorophille” to “Chlorophyll” and “activity photosynthetic” to “Photosynthetic activity”.

The figure 2 has been modified.

  1. Figure 3: It is not relevant to provide a figure to confirm one statement in one sentence. I recommend removing this figure.

It is not a confirmation. Figure 3 shows the high gelatinization of the DSE. The authors believe that the figure can remain. All the figures in a manuscript allude to what is being written.

  1. L233: “harvesting”, do you mean “cultivation”?

The word has been modified.

  1. L241: “Rise”, do you mean “Rice”?

it is indeed rice. The sentence has been modified.

  1. L362-363: “of endophyte fungi such as mycorrhizal fungi”, are they endophyte fungi?

The term “endophyte” has been removed

  1. L367: Please change “water available” to “available water”.

The sentence has been modified.

  1. L383: Please provide the meaning of “HMA”, do you mean AMF? The same for “PAC” (L396).

It has been rectified

  1. Table 1: Please add another column after the “host plant” column for “the main induced effect”, and if the terms “DSE” and “AMF” mean “DSE or AMF consortium”, please specify it. In the same table why “AM” is different from “AMF”?

Table 1 shows the different compatibility studies carried out by different researchers, and basically shows the variations of populations according to stress conditions. It will not add more information and would be repetitive.21. L441: please change “show” to “showed”.

Errors have been eliminated

Reviewer 4 Report

The manuscript entitled as “Importance of dark septate endophytes in agriculture in the face of climate change” reviews the effect of dark septate endophytes (DSEs) and the subjacent mechanisms that will help plants develop a higher tolerance to climate change. The topic is quite interesting since the climate change is a big challenge for agriculture since it affects crop productivity and yield.

The review topic is quite interesting and useful for readers and crop breeders to enhance their crop growth and productivity but there are lot of spelling and grammar mistakes throughout the text and authors have to be really carefully to revise the whole manuscript. And because of the presentation of words in a wrong way have changed the meaning of the scientific issues that authors actually wanted to highlight.

So I have some major concerns and authors can address these points and after that manuscript can be accepted for publication.

Line 102-Line 104 “conditions and activate the expression of hormone-regulated genes [43] (Li et al. 2022).  Higher accumulation of SOD has also been observed in wheat, rice [43,44] Li et al. 2022; Pang et al. 2020), and citric plants [45] (Sadeghi et al. 2020) inoculated with Penicillium”. The citation of references is wrong. So please carefully check it and correct it according to the Journal format.

Please check the Figure 2. The spelling of chlorophyll is wrong. Please also correct Activity photosynthetic to Photosynthetic activity.

Line 129 “[49,50]” There should be a space between the two reference numbers especially after commas. Author have this kind of mistake in many places. Please carefully check it.

Line 132-136 “Therefore, the use of DSEs Alternaria alternata, Paraphoma pye, and Paraphoma radicina on wheat and rice crops led to an increase in plant height, leaf growth, chlorophyll content, and photosynthetic rate of those plants, as well as a decrease in intercellular carbon dioxide, which alleviated the damage caused to photosynthetic processes by drought [43,44]”. The sentence is not clear. Please rewrite it again.

Line 136-138 “On the other hand, the inoculation of Ormosia hosiei with Acrocalymma vagum resulted in a damage-free root cell structure and an increase in the amount of chlorophyll and carotenoids produced [51,52]”. This sentence is grammatically wrong and it can be “On the other hand, the inoculation of Ormosia hosiei with Acrocalymma vagum resulted in a damage-free root cell structure and an increase in the amount of chlorophyll and carotenoids production [51, 52].

Line 143 “[45,54,55]”. Please provide the space after commas between reference numbers.

Line 169-170 “The low efficiency and prolonged use of fertilisation has contributed to making the problem worse [59]”. This sentence can be written as “The low efficiency and prolonged use of fertilisation has contributed to make this problem worse [59]”.

Line 241 “Rise has also shown that some”.

Please correct it. It is rice.

Line 244 to 245 “which suggests that the species used had the ability to improve the absorption of phosphorus from inorganic sources [89]”. Please rewrite it such as “which suggests that the species that were used, had the ability to improve the absorption of phosphorus from inorganic sources [89]”.

Conclusions

I suggest at the start of conclusion authors should not use the abbreviation i.e., Line 463-465 “DSEs have shown a capacity for mitigating and reducing the harmful effects of climate change in agriculture, such as salinity, drought, and reduced nutrient availability in the soil”.

Please also rewrite this sentence. It is grammatically wrong.

Line 471-472 “On the other hand, conducting studies is essential to allow the development of DSE formulates at an industrial scale to be used in agriculture in the future”. Please rewrite it.

Line 102-Line 104 “conditions and activate the expression of hormone-regulated genes [43] (Li et al. 2022).  Higher accumulation of SOD has also been observed in wheat, rice [43,44] Li et al. 2022; Pang et al. 2020), and citric plants [45] (Sadeghi et al. 2020) inoculated with Penicillium”. The citation of references is wrong. So please carefully check it and correct it according to the Journal format.

Please check the Figure 2. The spelling of chlorophyll is wrong. Please also correct Activity photosynthetic to Photosynthetic activity.

Line 129 “[49,50]” There should be a space between the two reference numbers especially after commas. Author have this kind of mistake in many places. Please carefully check it.

Line 132-136 “Therefore, the use of DSEs Alternaria alternata, Paraphoma pye, and Paraphoma radicina on wheat and rice crops led to an increase in plant height, leaf growth, chlorophyll content, and photosynthetic rate of those plants, as well as a decrease in intercellular carbon dioxide, which alleviated the damage caused to photosynthetic processes by drought [43,44]”. The sentence is not clear. Please rewrite it again.

Line 136-138 “On the other hand, the inoculation of Ormosia hosiei with Acrocalymma vagum resulted in a damage-free root cell structure and an increase in the amount of chlorophyll and carotenoids produced [51,52]”. This sentence is grammatically wrong and it can be “On the other hand, the inoculation of Ormosia hosiei with Acrocalymma vagum resulted in a damage-free root cell structure and an increase in the amount of chlorophyll and carotenoids production [51, 52].

Line 143 “[45,54,55]”. Please provide the space after commas between reference numbers.

Line 169-170 “The low efficiency and prolonged use of fertilisation has contributed to making the problem worse [59]”. This sentence can be written as “The low efficiency and prolonged use of fertilisation has contributed to make this problem worse [59]”.

Line 241 “Rise has also shown that some”.

Please correct it. It is rice.

Line 244 to 245 “which suggests that the species used had the ability to improve the absorption of phosphorus from inorganic sources [89]”. Please rewrite it such as “which suggests that the species that were used, had the ability to improve the absorption of phosphorus from inorganic sources [89]”.

Conclusions

I suggest at the start of conclusion authors should not use the abbreviation i.e., Line 463-465 “DSEs have shown a capacity for mitigating and reducing the harmful effects of climate change in agriculture, such as salinity, drought, and reduced nutrient availability in the soil”.

Please also rewrite this sentence. It is grammatically wrong.

Line 471-472 “On the other hand, conducting studies is essential to allow the development of DSE formulates at an industrial scale to be used in agriculture in the future”. Please rewrite it.

Author Response

RESPONSE TO REVIEW 4

Thank you very much for all the recommendations provided. The manuscript has been profoundly changed and the suggestions have been incorporated.

Line 102-Line 104 “conditions and activate the expression of hormone-regulated genes [43] (Li et al. 2022).  Higher accumulation of SOD has also been observed in wheat, rice [43,44] Li et al. 2022; Pang et al. 2020), and citric plants [45] (Sadeghi et al. 2020) inoculated with Penicillium”. The citation of references is wrong. So please carefully check it and correct it according to the Journal format.

The error has been removed.

Please check the Figure 2. The spelling of chlorophyll is wrong. Please also correct Activity photosynthetic to Photosynthetic activity.

The figure 2 has been modified and an explanation has been included in the legend of figure 2

Line 129 “[49,50]” There should be a space between the two reference numbers especially after commas. Author have this kind of mistake in many places. Please carefully check it.

References have been cited according to Jof's author guidelines:

References should be numbered in order of appearance and indicated by a numeral or numerals in square brackets—e.g., [1] or [2,3], or [4–6]. See the end of the document for further details on references.

Line 132-136 “Therefore, the use of DSEs Alternaria alternataParaphoma pye, and Paraphoma radicina on wheat and rice crops led to an increase in plant height, leaf growth, chlorophyll content, and photosynthetic rate of those plants, as well as a decrease in intercellular carbon dioxide, which alleviated the damage caused to photosynthetic processes by drought [43,44]”. The sentence is not clear. Please rewrite it again.

The sentence has been modified:  Therefore, the use of DSEs Alternaria alternata, Paraphoma pye, and Paraphoma radicina on wheat and rice crops led to an increase in the plant height, leaf growth, chlorophyll content, and photosynthetic rate, as well as a decrease in intercellular carbon dioxide, which alleviated the damage caused to photosynthetic processes by drought [43,44].

Line 136-138 “On the other hand, the inoculation of Ormosia hosiei with Acrocalymma vagum resulted in a damage-free root cell structure and an increase in the amount of chlorophyll and carotenoids produced [51,52]”. This sentence is grammatically wrong and it can be “On the other hand, the inoculation of Ormosia hosiei with Acrocalymma vagum resulted in a damage-free root cell structure and an increase in the amount of chlorophyll and carotenoids production [51, 52].

The sentence has been modified.

Line 143 “[45,54,55]”. Please provide the space after commas between reference numbers.

References have been cited according to Jof's author guidelines:

References should be numbered in order of appearance and indicated by a numeral or numerals in square brackets—e.g., [1] or [2,3], or [4–6]. See the end of the document for further details on references.

Line 169-170 “The low efficiency and prolonged use of fertilisation has contributed to making the problem worse [59]”. This sentence can be written as “The low efficiency and prolonged use of fertilisation has contributed to make this problem worse [59]”.

The sentence has been modified.  English has been reviewed by native English experts

Line 241 “Rise has also shown that some”.

Please correct it. It is rice.

The sentence has been modified.  English has been reviewed by native English experts

Line 244 to 245 “which suggests that the species used had the ability to improve the absorption of phosphorus from inorganic sources [89]”. Please rewrite it such as “which suggests that the species that were used, had the ability to improve the absorption of phosphorus from inorganic sources [89]”.

The sentence has been modified.  English has been reviewed by native English experts

Conclusions

I suggest at the start of conclusion authors should not use the abbreviation i.e., Line 463-465 “DSEs have shown a capacity for mitigating and reducing the harmful effects of climate change in agriculture, such as salinity, drought, and reduced nutrient availability in the soil”.

Please also rewrite this sentence. It is grammatically wrong.

The sentence has been modified.  English has been reviewed by native English experts

Line 471-472 “On the other hand, conducting studies is essential to allow the development of DSE formulates at an industrial scale to be used in agriculture in the future”. Please rewrite it.

The sentence has been modified.  English has been reviewed by native English experts

Round 2

Reviewer 3 Report

The authors satisfied all raised comments. I endorse the publication of the current manuscript.

The authors satisfied all raised comments. I endorse the publication of the current manuscript.

Reviewer 4 Report

The manuscript entitled as “Importance of dark septate endophytes in agriculture in the face of climate change” reviews the effect of dark septate endophytes (DSEs) and the subjacent mechanisms that will help plants develop a higher tolerance to climate change. The topic is quite interesting since the climate change is a big challenge for agriculture since it affects crop productivity and yield.

The article is revised carefully and all the comments have been considered and changes are made in the manuscript. According to points that have been raised in the old version, the current manuscript is updated very carefully and I am satisfied with the revisions. It can be accepted for publication. I have no further queries.